# Radio Frequency Ray Tracing via Stochastic Geometry

## Abstract

Radio frequency (RF) propagation modeling is essential for the design, analysis, and optimization of modern wireless sensing and communication systems. However, accurately modeling RF propagation in electrically large and complex environments remains a long-standing challenge, owing to the intricate interactions between RF signals and surrounding objects (e.g., reflection, diffraction, and scattering). Unlike conventional ray-tracing pipelines that hand-engineer interaction rules, or black-box neural surrogates that do not explicitly model physical structure, we introduce RFSG, a novel framework that integrates neural representations with physics-based RF propagation modeling. Starting with a stochastic representation of objects via random indicator functions, we derive the attenuation coefficient as a functional of the probability distributions of the underlying indicator functions under an exponential transport model. This formulation inherently satisfies key physical constraints such as reciprocity and reversibility. Building on this foundation, we employ object-centric neural representations to capture complex RF–object interactions while preserving the composability of traditional ray tracing. Extensive elevations on real-world testbeds demonstrate that RFSG consistently outperforms state-of-the-art neural baselines in prediction accuracy, while requiring significantly fewer training samples.

## 1 Introduction

Modern communication and sensing increasingly rely on wireless technologies that exploit electromagnetic (EM) waves for information exchange, fueling rapid advancements in mobile devices, automotive systems, and Internet-of-Things applications (De Alwis et al., 2021). Central to these developments is *radio frequency (RF) signal propagation modeling*, whose accurate characterization is critical not only for optimizing the deployment of wireless network infrastructures (Wei et al., 2017) and designing spatial protection zones for spectrum sharing (Testolina et al., 2024), but also for enhancing the performance of wireless sensing applications, including object detection, localization, and imaging (Hu et al., 2023; Zhao et al., 2022; Vakalis et al., 2019). Although EM wave propagation is fundamentally governed by Maxwell's equations (Haus & Melcher, 1989), obtaining analytical or numerical solutions in real-world environments is notoriously difficult due to the intricate geometries and nontrivial boundary conditions. As a result, accurately modeling RF propagation in electrically large and structurally complex environments remains an open challenge.

To address this challenge, ray tracing algorithms have been extensively adopted in the wireless industry for environments with known 3D scene models (Remcom, 2024; Aoudia et al., 2025; Chen & Zhang, 2023). These methods represent EM waves as a dense collection of rays emitted from a transmitter (Tx), which interact with surrounding objects and are subsequently captured by receivers (Rx). Despite their wide adoption, conventional RF ray tracing approaches assume fully known and deterministic environments. In contrast, stochastic geometry offers a probabilistic framework that naturally accounts for epistemic uncertainties, such as unknown node positions or random spatial configurations, by modeling them as spatial point processes.

Recent advances in neural rendering, including Neural Radiance Fields (Mildenhall et al., 2021) and 3D Gaussian Splatting (Kerbl et al., 2023), have demonstrated remarkable capabilities in capturing complex light transport for photorealistic scene reconstruction. These developments suggest promising avenues for RF propagation modeling, as both domains share underlying physical principles. Data-driven approaches, such as NeRF$^2$ (Zhao et al., 2023), NeWRF (Lu et al., 2024), RFcanvas

(Chen et al., 2024) and WRF-GS (Wen et al., 2025), have shown potential in learning fine-grained scene details and RF-object interactions directly from measurements. Nevertheless, these methods often trade off the flexibility of conventional ray tracing and suffer from severe data inefficiency, requiring roughly 200 channel samples per square foot to achieve satisfactory performance.

In this paper, we propose **Radio Frequency Stochastic Geometry** (RFSG), a novel framework that synergistically combines the high fidelity of neural representations with the flexibility and interpretability of ray tracing. We introduce a stochastic geometric model for scenes comprising unknown 3D objects and model each RF signal propagation path between the transmitter and receiver as a sequence of propagation segments, each representing either free-space propagation or object-inside propagation until encountering a boundary. The object-inside propagation distance is modeled as an *exponential* random variable (Miller et al., 2024), with a rate parameter that is related to the corresponding material properties. We derive conditions for *exponential* transport and establish sufficient functional relationships among propagation parameters (e.g., attenuation), material properties, and stochastic geometry by enforcing physical constraints such as reciprocity. Following (Miller et al., 2024; Heitz et al., 2015; Jakob et al., 2010), we further incorporate anisotropic effects and extend the framework to handle commonly used geometric representations, such as implicit surfaces. By leveraging neural representations to implicitly capture fine structural details and complex material properties, our method enables differentiable ray-geometry intersections, facilitating gradient-based optimization.

Our contributions can be summarized as follows:

- We propose a novel framework that learns dedicated neural representations for individual objects in a scene and seamlessly integrates them into a physically interpretable ray tracing pipeline.

- We model RF signal propagation between the transmitter and receiver as a sequence of exponential transport processes, and derive functional relationships linking attenuation, material properties, and stochastic object geometry under physically grounded constraints.

- We evaluate extensively across diverse real-world wireless environments. Our results demonstrate 's effectiveness and its potential to foster real-world wireless applications.

## 2 BACKGROUND AND PRELIMINARIES

In this section, we provide background knowledge on RF ray tracing and wireless channel.

**RF Ray Tracing.** Ray tracing has emerged as a powerful technique for simulating radio wave propagation, driven by the rapid expansion of wireless communication and sensing applications. While full-wave EM solvers offer high-fidelity predictions, their prohibitive computational cost renders them impractical for large and electrically complex environments. In contrast, RF ray tracing leverages the principles of geometric optics to approximate EM wave behavior, providing a favorable trade-off between accuracy and computational efficiency. In this paradigm, RF waves are modeled as rays that interact with the environment through reflection, refraction, diffraction, and scattering. Although these interactions are fundamentally governed by Maxwell's equations, they can be well-approximated by ray optics when the carrier wavelength is significantly smaller than the characteristic dimensions of environmental features. The ray tracing process begins with environment modeling, where the physical scene is represented as a collection of object geometry, each annotated with electromagnetic material parameters such as relative permittivity and conductivity. From each transmitter location, rays are launched in discrete angular directions with a prescribed spatial density. Ray–surface intersections are then computed, and secondary rays are generated according to the number and types of interactions encountered. Depending on the interaction type — such as reflection, refraction, or diffraction —— ray trajectories are updated following geometric optics principles. At each interaction, the directional radiance (i.e., the power carried along a specific direction) is evaluated as a function of the material's electromagnetic properties. Ray propagation continues until a ray reaches the receiver or exceeds a predefined maximum number of interactions. Finally, all rays arriving at the Rx are coherently superimposed, with relative phases determined by their respective path lengths, to reconstruct the received signal.

**Wireless Channel.** A generic wireless communication system typically consists of a TX that generates and modulates an information signal, which is then transmitted through the wireless channel to an RX. The transmitted signal can be represented as a complex-valued number $s = Ae^{j\varphi}$,

where $A$ denotes amplitude and $\varphi$ denotes phase respectively. During propagation, its amplitude is degraded by an attenuation factor $A_{\text{att}}$, and the phase is rotated by $\Delta\varphi$. In the case of *free-space* propagation —— where no obstacles obstruct the line-of-sight path — the amplitude attenuation is inversely proportional to the propagation distance, and the phase rotation is linearly inversely proportional to the distance (Molisch, 2012):

$$a(r) = \frac{c}{4\pi f r}, \ \ \Delta\varphi(r) = -\frac{2\pi f r}{c}, \tag{1}$$

where $r$ is the propagation distance, $f$ is the carrier frequency, and $c$ is the speed of light. In realistic wireless environments, multiple propagation mechanisms—including reflection, scattering, refraction, and diffraction—give rise to multiple delayed and distorted copies of the transmitted signal. Consequently, the received signal can be expressed as (Tse & Viswanath, 2005)

$$y = Ae^{j\varphi}\sum_{l=1}^{L}T_l e^{j\Delta\varphi_l}, \tag{2}$$

where $L$ is the total number of propagation paths, $T_l$ accounts for the attenuation for path $l$, and $\Delta\varphi_l$ is the corresponding phase rotation. The wireless channel $h$, is defined as the ratio of the received signal and transmitted signal, capturing the cumulative effect of the propagation environment:

$$h = \frac{y}{s} = \sum_{l=1}^{L}T_l e^{j\Delta\varphi_l}. \tag{3}$$

## 3 RELATED WORK

**RF propagation and EM field Modeling.** Full-wave electromagnetic simulation methods, such as the Finite Element Method (FEM) (Jin, 2015) and the Finite-Difference Time-Domain (FDTD) method (Taflove et al., 2005), achieve high accuracy but incur prohibitive computational costs, restricting their use to small-scale RF device designs such as antenna prototyping. For large-scale scenarios, ray-tracing-based approaches (Ling et al., 1989; Oestges et al., 2003; Remcom, 2024) offer higher efficiency by approximating electromagnetic wave propagation through the principles of geometrical optics. However, their accuracy critically depends on precise knowledge of object geometries and material properties—requirements that exceed the sensing capabilities of conventional measurement systems. Even in the case of specular objects with simple geometries, visual 3D reconstruction techniques often fail to recover microscale surface details that, while visually negligible, can substantially alter RF wave interactions.

Recent advances in scene representations have established a significant position in computer vision and computer graphics, particularly following the emergence of NeRF (Mildenhall et al., 2021) and 3D Gaussian Splatting (Kerbl et al., 2023). Several studies have adapted neural representations and 3D Gaussian models for RF applications, with NeRF$^2$ (Zhao et al., 2023), (Lu et al., 2024) and (Wen et al., 2025) as being notable examples. These methods represent the entire scene as either an implicit neural function or a collection of analytical 3D Gaussian kernels, which hinders explicit boundary delineation and the isolation of object-specific RF interaction properties. This limitation reduces adaptability to diverse RF environments and typically requires a larger volume of training data. WiNeRT (Orekondy et al., 2023) mitigates some of these issues by incorporating neural material reflection parameters into ray-tracing-based simulations, yet its reliance on traditional 3D geometry from CAD model input remains insufficient for capturing detailed structural features of physical objects (Seyb et al., 2024). More recently, RFscape (Chen et al., 2025) leverages a learnable signed distance function (SDF) alongside a neural material network to jointly model complex geometries and intrinsic RF properties and embedd them into ray tracing pipeline. However, it overlooks fundamental physical constraints —— such as reciprocity and reversibility —— in ray propagation, limiting its physical fidelity in practical applications.

**Exponential Transport for Stochastic Object Geometry.** Previous work has already explored exponential transport models in the context of stochastic geometry. For example, Mishchenko et al. (2006) studied exponential transport for stochastic microparticle distributions, which can be formalized as a Poisson Boolean model of stochastic geometry (d'Eon, 2018; Jarabo et al., 2018), where particle locations are independent and modeled as a spatial Poisson process (Chiu et al., 2013). Within this framework, the attenuation coefficient can be derived analytically from the probability

distributions governing particle location, size, material, shape, and orientation (Heitz et al., 2015; Jakob et al., 2010). In contrast, a separate line of research has investigated how to convert deterministic surface representations (e.g., SDF) into volumetric forms that approximately preserve geometric behavior (Oechsle et al., 2021; Sellán & Jacobson, 2022; Wang et al., 2021; Yariv et al., 2021). While empirically successful, such approaches remain heuristic, as they require making ad hoc choices about which properties of the surface representation to preserve in the volumetric domain. More recently, Miller et al. (2024) derived volumetric representations of opaque solids directly from the axioms of exponential transport using stochastic geometry theory, providing a principled explanation for why volumetric neural rendering methods are able to recover solid geometry.

Building upon these insights, we advance the theory of exponential transport from optics into the RF domain, where several fundamental differences make the problem significantly more challenging. Unlike visible light, RF signals possess much longer wavelengths, which enable them to penetrate common materials. This distinction fundamentally invalidates the assumption in Miller et al. (2024) that rays are terminated at opaque solid boundaries, since in RF propagation, the dominant effect is cumulative attenuation within the objects. Moreover, attenuation in RF propagation is inherently material-dependent: different media (e.g., concrete, wood, metal) induce distinct propagation behaviors governed by material property such as permittivity, conductivity. In contrast, the formulation of Miller et al. (2024) is agnostic to material properties and is developed in the context of optical transport, and therefore does not capture the physics of RF propagation. By explicitly incorporating object penetration and material-dependent attenuation representations, our formulation advances exponential transport into the RF regime, providing the first stochastic geometry framework that rigorously explains how RF signals propagate and interact with complex environments.

## 4 METHODOLOGY

In this section, we establish a comprehensive framework of exponential transport tailored for RF ray tracing, bridging theoretical foundations with neural modeling capabilities. Section 4.1 introduces the notations and formal definitions of exponential transport under stochastic object geometry. Section 4.2 presents the conditions under which exponential transport holds and derives physically consistent attenuation representations, including reciprocity constraints. Formulations are extended to accommodate anisotropic property and implicit surface parameterizations. Section 4.3 demonstrates how the resulting expressions can be integrated into RF ray tracing with neural representations. Finally, Section 4.4 details the optimization procedure for our proposed approach.

### 4.1 DEFINITIONS AND NOTATIONS

To set the stage for rigorous analysis of exponential transport in RF ray tracing, we first introduce some notations and definitions for stochastic geometry and object-inside propagation, similar to those used in (Miller et al., 2024).

**Notations.** We use $\boldsymbol{r}_{\boldsymbol{x},\boldsymbol{w}}(t) \triangleq \boldsymbol{x} + t \cdot \boldsymbol{w}$ to denote the point on a ray with origin $\boldsymbol{x} \in \mathbb{R}^3$ and direction $\boldsymbol{\omega} \in \mathcal{S}^2$ after propagating distance $t \in [0, \infty)$. We denote $d^*_{\boldsymbol{x},\boldsymbol{\omega}}$ as the object-inside propagation distance and $\boldsymbol{r}_{\boldsymbol{x},\boldsymbol{w}}(d^*_{\boldsymbol{x},\boldsymbol{\omega}})$ as the intersection point between object and free-space.

**Definition 1.** Define a indicator function $I : \mathbb{R}^3 \to \{0, 1\}$ as a binary scalar field, and associate it with a object geometry $\mathcal{O} \triangleq \{\boldsymbol{x} \in \mathbb{R}^3 : I(\boldsymbol{x}) = 1\}$.

**Definition 2.** When the indicator function $I(\boldsymbol{x})$ is a random scalar field, the associated $\mathcal{O}$ is called a stochastic object geometry, for which we define the occupancy $o : \mathbb{R}^3 \to [0, 1]$ and vacancy $v : \mathbb{R}^3 \to [0, 1]$:

$$
\begin{aligned}
o(\boldsymbol{x}) &= \Pr(I(\boldsymbol{x}) = 1), \\
v(\boldsymbol{x}) &= \Pr(I(\boldsymbol{x}) = 0) = 1 - o(\boldsymbol{x}).
\end{aligned}
\tag{4}
$$

**Definition 3.** In a scene with stochastic object geometry $\mathcal{O}$, define the tail distribution of the object-inside propagation distance $d^*_{\boldsymbol{x},\boldsymbol{\omega}}$:

$$
T_{\boldsymbol{x},\boldsymbol{\omega}}(t) \triangleq \Pr\{d^*_{\boldsymbol{x},\boldsymbol{\omega}} \geq t\}.
\tag{5}
$$

The object-inside propagation distribution is the probability density function (pdf) of object-inside propagation distance $d^*_{\boldsymbol{x},\boldsymbol{\omega}}$:

$$
p_{\boldsymbol{x},\boldsymbol{\omega}}(t) = \frac{\mathrm{d}(1 - T_{\boldsymbol{x},\boldsymbol{\omega}}(t))}{\mathrm{d}t} = -\frac{\mathrm{d}T_{\boldsymbol{x},\boldsymbol{\omega}}(t)}{\mathrm{d}t}.
\tag{6}
$$

The *absolute value/magnitude* of attenuation coefficient $a(\boldsymbol{x}, \boldsymbol{\omega})$ at point $\boldsymbol{x}$ and direction $\boldsymbol{\omega}$ within object is the pdf value of zero object-inside propagation distance:

$$|a(\boldsymbol{x}, \boldsymbol{\omega})| = \sigma(\boldsymbol{x}, \boldsymbol{\omega}) = p_{\boldsymbol{x}, \boldsymbol{\omega}}(0). \tag{7}$$

$T_{\boldsymbol{x}, \boldsymbol{\omega}}$ inherits the following properties from ray propagation: (*i*) *monotonically non-increasing* since $T_{\boldsymbol{x}, \boldsymbol{\omega}}(t) \leq T_{\boldsymbol{x}, \boldsymbol{\omega}}(s)$ if $t > s$; (*ii*) $T_{\boldsymbol{x}, \boldsymbol{\omega}}(0) = 1$. In this work, we restrict our attention to the magnitude component of attenuation coefficient, i.e., $\sigma(\boldsymbol{x}, \boldsymbol{\omega}) = |a(\boldsymbol{x}, \boldsymbol{\omega})|$, while leaving the treatment of the phase component, $\varphi(\boldsymbol{x}) = \angle a(\boldsymbol{x}, \boldsymbol{\omega})$, to future investigation.

### 4.2 Exponential Transport for RF Ray Propagation

Similar to most commonly in computer vision and computer graphics (Preisendorfer, 2014), we model the object-inside propagation distance as an exponential random variable. Then, Equations 5 to 7 imply:

$$T_{\boldsymbol{x}, \boldsymbol{\omega}}(t) = \exp(-\int_0^t \sigma(\boldsymbol{r}_{\boldsymbol{x}, \boldsymbol{\omega}}(s), \boldsymbol{\omega}) \mathrm{d}s), \tag{8}$$

$$p_{\boldsymbol{x}, \boldsymbol{\omega}}(t) = \sigma(\boldsymbol{r}_{\boldsymbol{x}, \boldsymbol{\omega}}(t), \boldsymbol{\omega}) T_{\boldsymbol{x}, \boldsymbol{\omega}}(t). \tag{9}$$

Therefore, the magnitude of attenuation coefficient stands for the exponential rate parameter of the single-medium propagation distance.

Each propagation path of the RF signals between the transmitter and the receiver is modeled as a sequence of propagation segments, where each segment corresponds either to free-space propagation or to propagation inside an object until reaching a boundary. Right after the intersection, we consider reflective (including Specular reflection and diffuse reflection) and penetrative rays for the next ray propagation, which also implies for later intersection points. Before stating our main theoretical results, we emphasis the physical plausibility for ray tracing of RF signals:

(*i*) **Basic reciprocity.** Consider two points $\boldsymbol{x}, \boldsymbol{y}$ in free space, $T_{\boldsymbol{x}, \boldsymbol{\omega}}(t) = T_{\boldsymbol{y}, -\boldsymbol{\omega}}(t)$ if $\boldsymbol{y} = \boldsymbol{r}_{\boldsymbol{x}, \boldsymbol{\omega}}(t)$.

(*ii*) **Reflection reciprocity.** Consider two points $\boldsymbol{x}$ and $\boldsymbol{z}$ in free-space, connected by a single reflection at a point on the boundary $\boldsymbol{y}$. Let the incident ray from $\boldsymbol{x}$ to $\boldsymbol{y}$ have direction $\boldsymbol{\omega}$, and the reflected direction from $\boldsymbol{y}$ to $\boldsymbol{z}$ be $\boldsymbol{\omega}'$ (for specular reflection, $\boldsymbol{\omega}'$ can be computed via Snell's Law (Balanis, 2012) as $\boldsymbol{\omega}' = \boldsymbol{\omega} - 2(\boldsymbol{\omega} \cdot \boldsymbol{n_y})\boldsymbol{n_y}$, where $\boldsymbol{n_y}$ is the normal vector at $\boldsymbol{y}$). Then, we must have:

$$T_{\boldsymbol{x}, \boldsymbol{\omega}}(\|\boldsymbol{y} - \boldsymbol{x}\|) \cdot T_{\boldsymbol{y}, \boldsymbol{\omega}'}(\|\boldsymbol{z} - \boldsymbol{y}\|) = T_{\boldsymbol{z}, -\boldsymbol{\omega}'}(\|\boldsymbol{y} - \boldsymbol{z}\|) \cdot T_{\boldsymbol{y}, -\boldsymbol{\omega}}(\|\boldsymbol{x} - \boldsymbol{y}\|). \tag{10}$$

(*iii*) **Penetration reciprocity.** Consider a ray traveling from point $\boldsymbol{x}$ in free space to point $\boldsymbol{v}$ in free-space, passing through a single-medium object along direction $\boldsymbol{\omega}$. Let $\boldsymbol{y}$ and $\boldsymbol{z}$ denote the two ordered intersection points where the ray enters and exits the object, respectively. Then, we must have:

$$T_{\boldsymbol{x}, \boldsymbol{\omega}}(\|\boldsymbol{y} - \boldsymbol{x}\|) \cdot T_{\boldsymbol{y}, \boldsymbol{\omega}}(\|\boldsymbol{z} - \boldsymbol{y}\|) \cdot T_{\boldsymbol{z}, \boldsymbol{\omega}}(\|\boldsymbol{v} - \boldsymbol{z}\|)$$
$$= T_{\boldsymbol{v}, -\boldsymbol{\omega}}(\|\boldsymbol{z} - \boldsymbol{v}\|) \cdot T_{\boldsymbol{z}, -\boldsymbol{\omega}}(\|\boldsymbol{y} - \boldsymbol{z}\|) \cdot T_{\boldsymbol{y}, -\boldsymbol{\omega}}(\|\boldsymbol{x} - \boldsymbol{y}\|). \tag{11}$$

(*iv*) **Reversibility.** Object geometry inferred from the underlying indicator function $I$ is consistent when evaluated along any ray, regardless of whether the ray direction $\boldsymbol{\omega}$ is forward or reversed.

Inspired by Theorem 4 in (Miller et al., 2024), we further extend and generalize the exponential transport for RF ray tracing, which takes the combination of (multiple) reflective surfaces and (multiple) penetration through objects into consideration.

**Theorem 1.** *Consider a random indicator function $I$ and associated stochastic object geometry $\mathcal{O}$. We assume that $\sigma(\boldsymbol{x}, \boldsymbol{\omega})$ is differentiable with respect to point $\boldsymbol{x}$, with its gradient bounded for any $\boldsymbol{\omega} \in \mathcal{S}^2$. Then, for any ray with origin $\boldsymbol{x}$ in a certain object, direction $\boldsymbol{\omega}$, the distribution of the object-inside propagation distance $p_{\boldsymbol{x}, \boldsymbol{\omega}}$ is exponential if and only if $I_{\boldsymbol{x}, \boldsymbol{\omega}}$, constitutes a continuous-time discrete-state Markov process. Formally, this condition requires that for any $t$:*

$$\Pr\{I_{\boldsymbol{x}, \boldsymbol{\omega}}(t) \mid I_{\boldsymbol{x}, \boldsymbol{\omega}}(t_n), t_n < t, n = 1, \ldots, N\} = \Pr\{I_{\boldsymbol{x}, \boldsymbol{\omega}}(t) \mid I_{\boldsymbol{x}, \boldsymbol{\omega}}(\max_n t_n)\}. \tag{12}$$

*Furthermore, the process is physically plausible (i.e. satisfying reciprocity and reversibility) if the attenuation magnitude $\sigma_\delta(\boldsymbol{x}, \boldsymbol{\omega})$ satisfies:*

$$\sigma_\delta(\boldsymbol{x}, \boldsymbol{\omega}) = \frac{(m(\boldsymbol{x})(1 - v(\boldsymbol{x})) + 1)|\boldsymbol{\omega} \cdot \nabla v(\boldsymbol{x})|}{1 - v(\boldsymbol{x})}. \tag{13}$$

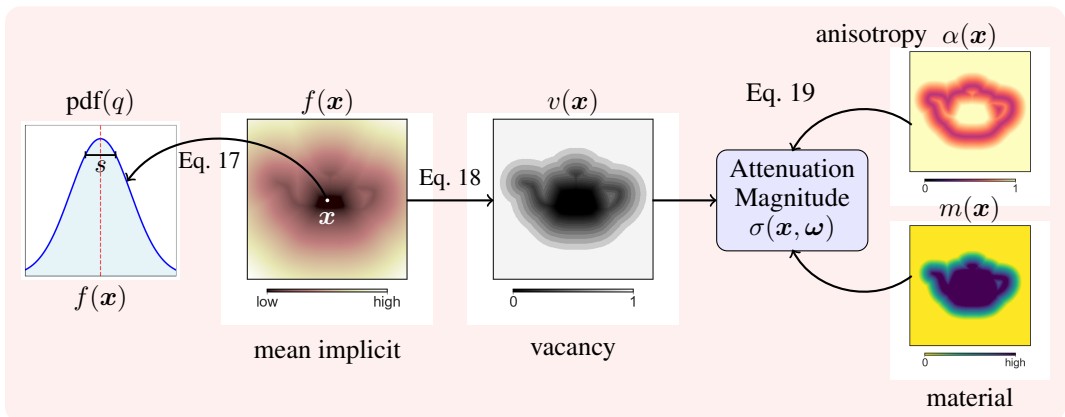

Figure 1: Overview of Subsection 4.2.

*Remark.* In Theorem 1, we analyze the attenuation magnitude within objects. We further derive an attenuation representation that is valid at any point in free space,

$$\sigma_\delta(\boldsymbol{x}, \boldsymbol{\omega}) = \frac{(m(\boldsymbol{x})v(\boldsymbol{x}) + 1)|\boldsymbol{\omega} \cdot \nabla v(\boldsymbol{x})|}{v(\boldsymbol{x})}. \tag{14}$$

In practice, we adopt the closed-form expression for free-space propagation (i.e., Eq. 1) rather than discretizing each ray and sampling along its path for computational efficiency.

We explain the notation $\sigma_\delta$ below, and provide the proof in Appendix A.

**Anisotropy.** In Theorem 1, $\sigma_\delta(\boldsymbol{x}, \boldsymbol{\omega})$ corresponds to the fully anisotropic case, i.e., the distribution of normals is a Dirac delta. Following the discussion of *anisotropy* in (Miller et al., 2024), we can further extend representation in Eq. 13 to a convex combination of two cases—the fully isotropic case (i.e., normals uniformly distributed) and the fully anisotropic case:

$$\sigma(\boldsymbol{x}, \boldsymbol{\omega}) = (m(\boldsymbol{x})(1 - v(\boldsymbol{x})) + 1)\frac{\|\nabla v(\boldsymbol{x})\|}{1 - v(\boldsymbol{x})} \cdot \sigma_{\text{mix}}^\perp(\boldsymbol{x}, \boldsymbol{\omega}), \tag{15}$$

$$\sigma_{\text{mix}}^\perp = \alpha(\boldsymbol{x})|\boldsymbol{\omega} \cdot \boldsymbol{n}(\boldsymbol{x})| + \frac{1 - \alpha(\boldsymbol{x})}{2}, \tag{16}$$

where $\boldsymbol{n}(\boldsymbol{x}) = \frac{\nabla v(\boldsymbol{x})}{\|\nabla v(\boldsymbol{x})\|}$ is the unit normal of the level set of $v$ passing through $\boldsymbol{x}$, $\alpha(\boldsymbol{x}) \in [0, 1]$ represents the anisotropic parameter.

**Go beyond binary.** Definition 1 and Definition 2 define object geometry through the binary indicator function. In practice, it is often preferable to represent geometry via a non-binary scalar field, which can encode richer information (Osher et al., 2004). We consider a generalized definition of non-binary random field: an implicit function $G : \mathbb{R}^3 \to \mathbb{R}$ is defined as a random real scalar field with associated indicator $I(\boldsymbol{x}) = \mathbf{1}_{G(\boldsymbol{x}) \leq 0}$, we define CDF, PDF and mean of random field $G$:

$$\text{cdf}(q) \triangleq \Pr\{G(\boldsymbol{x}) \leq q\}, \ \ \text{pdf}(q) \triangleq \frac{\text{d}\,\text{cdf}(q)}{\text{d}q}, \ \ q \in \mathbb{R} \ \text{with} \ f(\boldsymbol{x}) \triangleq \mathbb{E}[G(\boldsymbol{x})]. \tag{17}$$

We follow Proposition 7 in (Miller et al., 2024). Consider a symmetric (unbiased) random field $G$,

$$v(\boldsymbol{x}) = \Pr(G(\boldsymbol{x} > 0) = \Psi(sf(\boldsymbol{x})), \ \ o(\boldsymbol{x})\Pr(G(\boldsymbol{x} \leq 0) = \Psi(-sf(\boldsymbol{x})), \tag{18}$$

we can further generalize the representation in Theorem 1:

$$\sigma(\boldsymbol{x}, \boldsymbol{\omega}) = (m(\boldsymbol{x})\Psi(-sf(\boldsymbol{x})) + 1)\frac{s\psi(sf(\boldsymbol{x}))\|\nabla f(\boldsymbol{x})\|}{\Psi(-sf(\boldsymbol{x}))} \cdot \sigma_{\text{mix}}^\perp(\boldsymbol{x}, \boldsymbol{\omega}), \tag{19}$$

with $\sigma_{\text{mix}}^\perp(\boldsymbol{x}, \boldsymbol{\omega})$ as defined in Eq. 16, where $\Psi(\cdot), \psi(\cdot)$ denotes the CDF and PDF of a certain symmetric distribution (e.g., Laplace, Gaussian, Logistic), $s > 0$ represents the scaling coefficient. We summarize our theoretical results in Fig. 1.

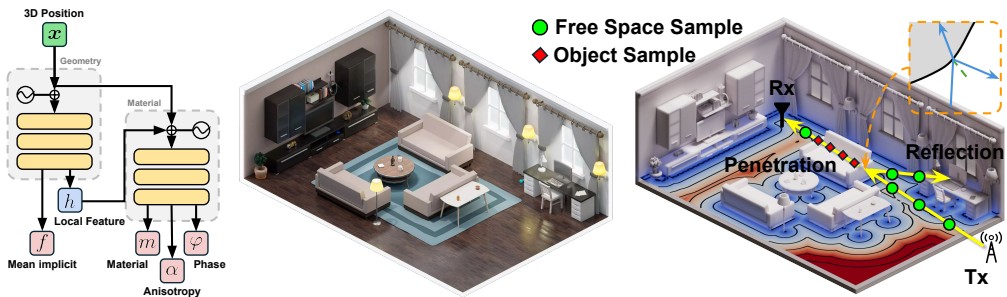

Figure 2: RF Ray Tracing with Neural Representation.

## 4.3 RF RAY TRACING WITH RFSG

To accurately describe stochastic objects, we model their geometry using the mean implicit function $f(\boldsymbol{x})$. Nevertheless, geometry alone is insufficient for calculating the complex-valued attenuation coefficient, which additionally depends on the local material properties of the medium $m(\boldsymbol{x})$, anisotropy parameter $\alpha(\boldsymbol{x})$ and phase $\varphi(\boldsymbol{x})$. Conventional RF ray tracing (Ling et al., 1989) computes Fresnel coefficients under the assumption of homogeneous and *known* materials. In practice, however, material properties are highly heterogeneous across both surfaces and interiors, and are often not measurable. To overcome this limitation, we parameterize these unknown parameters with a trainable neural network. Inspired by NeRF (Mildenhall et al., 2021), we adopt multi-layer perceptrons (MLPs) with the following mapping:

$$F_{\boldsymbol{\Theta}} : \boldsymbol{x} \mapsto \big(f(\boldsymbol{x}),\, m(\boldsymbol{x}),\, \alpha(\boldsymbol{x}),\, \varphi(\boldsymbol{x})\big). \tag{20}$$

The complex-valued attenuation coefficient is then given by

$$a(\boldsymbol{x}, \boldsymbol{\omega}) = \sigma(\boldsymbol{x}, \boldsymbol{\omega})\, e^{j\varphi(\boldsymbol{x})}, \tag{21}$$

where $\sigma(\boldsymbol{x}, \boldsymbol{\omega})$ follows equation 19 or equation 1. By precisely localizing ray–object interactions with vanishing error, our model has the potential to eliminate the sim-to-real discrepancies.

RF propagation along each path involves complex interactions with one or more objects. A key step in ray tracing is locating the intersection points, which in our approach is achieved by iteratively searching for the zero-crossing of $f(\boldsymbol{x})$ along the ray direction. The search terminates once $|f(\boldsymbol{x})| < \epsilon$ (with $\epsilon = 0.01$) or the maximum iteration/depth is reached. At each intersection, the local surface normal $\boldsymbol{n}(\boldsymbol{x})$ is estimated via finite differences:

$$\boldsymbol{n}(\boldsymbol{x}) = \left[ \frac{\frac{f(\boldsymbol{x}+\epsilon_x)-f(\boldsymbol{x}-\epsilon_x)}{2\epsilon_x}}{\left\| \frac{f(\boldsymbol{x}+\epsilon_x)-f(\boldsymbol{x}-\epsilon_x)}{2\epsilon_x} \right\|},\ \frac{\frac{f(\boldsymbol{x}+\epsilon_y)-f(\boldsymbol{x}-\epsilon_y)}{2\epsilon_y}}{\left\| \frac{f(\boldsymbol{x}+\epsilon_y)-f(\boldsymbol{x}-\epsilon_z)}{2\epsilon_y} \right\|},\ \frac{\frac{f(\boldsymbol{x}+\epsilon_z)-f(\boldsymbol{x}-\epsilon_z)}{2\epsilon_z}}{\left\| \frac{f(\boldsymbol{x}+\epsilon_z)-f(\boldsymbol{x}-\epsilon_z)}{2\epsilon_z} \right\|} \right]^{\top}, \tag{22}$$

where $\epsilon_x$, $\epsilon_y$ and $\epsilon_z$ are small perturbations in the $x$, $y$ and $z$ axes, respectively. The direction of the specular reflected ray is then computed by Snell's law, while penetrative rays preserve their incident direction. To control the complexity of ray tracing, we adopt a Monte Carlo strategy in which either of the reflective or penetrative ray is selected for further propagation, with probabilities proportional to their respective directional attenuation coefficients.

Following standard ray-tracing procedures, rays are emitted from the Tx, and their interactions with scene objects are recursively computed until valid paths reaching the Rx are identified. Each valid path $l$ is characterized by a transmittance $T_l = \prod_i \exp(-a_l^{(i)}) = \exp(-\sum_i a_l^{(i)})$ with $a_l^{(i)}$ denotes the total attenuation in segment $i$ along path $l$, and its traversal distance $\tau_l = \sum_i \tau_l^{(i)}$. When segment is inside object, $a_l^{(i)}$ is approximated by sampling along the ray with a small step size $\Delta t$:

$$a_l^{(i)} = \int_0^{\tau_l^{(i)}} a(\boldsymbol{p}_0 + t \cdot \boldsymbol{\omega}) \mathrm{d}t \approx \sum_{n=0}^{N_l^{(i)}-1} a(\boldsymbol{p}_0 + n\Delta t \cdot \boldsymbol{\omega}) \Delta t, \tag{23}$$

where $\boldsymbol{p}_0$ denotes ray origin and $N_l^{(i)} = \frac{\tau_l^{(i)}}{\Delta t}$ is the total number of samples along the ray segment. The (estimated) received signal is thus expressed as:

$$\hat{S}_{\text{Rx}} = S_{\text{Tx}} \cdot \sum_l T_l \cdot e^{-j2\pi f \tau_l}. \tag{24}$$

### 4.4 OPTIMIZATION

To faithfully capture both the geometry and material properties of the objects, and to accurately model RF signal propagation, we optimize the parameters $\Theta$ by minimizing the sim-to-real discrepancy, i.e., $\sum_{m=1}^{M} \mathcal{L}(S_{\text{Rx}}^{(m)}, \hat{S}_{\text{Rx}}^{(m)})$, where $S_{\text{Rx}}^{(m)}$ is $m$-th RF measurement in the scene, $M$ is the number of measurements, $\mathcal{L}$ is the loss that quantifies the discrepancy between real measurements and model prediction. To encourage smooth and physically consistent reconstructions, we additionally incorporate a discrete Laplacian regularization. The overall optimization objective becomes

$$\Theta^o = \arg\min_{\Theta} \sum_{m=1}^{M} \mathcal{L}(S_{\text{Rx}}^{(m)}, \hat{S}_{\text{Rx}}^{(m)}) + \lambda \sum_{\boldsymbol{x} \in \mathcal{X}} (\|\nabla^2 f(\boldsymbol{x})\| + \|\nabla^2 m(\boldsymbol{x})\| + \|\nabla^2 \alpha(\boldsymbol{x})\| + \|\nabla^2 \varphi(\boldsymbol{x})\|),$$

(25)

where $\mathcal{X}$ denotes the set of randomly sampled points $\boldsymbol{x}$ within the scene. We expect that only a small number of RF measurements (i.e., small $M$) is sufficient, since the governing physical laws of ray propagation provide strong structural constraints.

*Remark.* The signed distance function (SDF) can be regarded as a special case of $f(\boldsymbol{x})$. In this case, an additional eikonal regularization (Sethian, 1996) is required to enforce the unit-norm constraint on the SDF gradient.

## 5 EXPERIMENTS

### 5.1 EXPERIMENTAL SETTINGS

**Dataset.** We use an real-world open-source BLE dataset provided in NeRF[2] (Zhao et al., 2023), where the facility occupies $15000\,\text{ft}^2$. It contains 21 BLE gateways, which operate at 2.4 GHz to collect the ID and RSSI of BLE beacons. Each dataset item is a 21-dimensional tuple, including the RSSI values detected by 21 gateways, plus the position of the BLE node. The RSSI value is set to -100 dB by default if the gateway does not detect any signal. To conduct the experiments in a regular cuboid, we cropped the data set, that is, the full floor plan, to a subregion consisting of adjacent rooms. 80% and 20% of the dataset are chosen from training and testing datasets.

**Baseline.** We employ **NeRF**[2] (Zhao et al., 2023) as a baseline, as it represents the state-of-the-art in neural channel estimation, already shown superior performance to other deep learning based approaches such as Deep Convolutional Generative Adversarial Network (DCGAN) and Variational Autoencoder (VAE). Inspired by the results in RFCanvas (Chen et al., 2024) and WiNeRT (Orekondy et al., 2023), some traditional machine learning approaches may achieve surprisingly better performance. Thus, we introduce two more baselines, **$k$-NN**: Predicting the channel given the closest match to the input spatial coordinates in terms of Euclidean distance; **MRI** (Shin et al., 2014): Interpolating the RSSI values at the unsampled locations using a basic radio propagation model.

**Model Implementation.**

- Geometry Model: we employ an 8-layer Multilayer Perceptron (MLP) with a hidden dimension of 64 and integration of skip connections. The model takes 3D cooridinates as input, i.e. $G_{\Omega}: \boldsymbol{x} \rightarrow (f, \boldsymbol{h})$, where an additional 128-dimensional local features $\boldsymbol{h}$ is the output of the geometry model. To cope with the spectral bias inherent in coordinate-based MLP, the 3D coordinate input $\boldsymbol{x}$ undergoes positional encoding with 6 frequencies.

- Material Model: the material model consists of an 8-layer MLP with a width of 128 and incorporates skip connections, where the inputs are spatial coordinate $\boldsymbol{x}$ and feature $\boldsymbol{h}$ from geometry model. This architecture is designed to capture electromagnetic material properties.

**Training Configurations.** The training configuration includes a batch size of 4 and utilizes the Adam optimizer (Kingma & Ba, 2014). The learning rate is set at $1 \times 10^{-4}$. On a NVIDIA RTX 5090 GPU, the network typically converges after approximately $30k$ iterations.

### 5.2 RESULTS

Data-driven approaches have emerged as promising solutions for BLE localization, but they typically require large datasets for fingerprint matching or network training. For example, NeRF[2] necessitates roughly 200 channel measurements per square foot, posing significant practical challenges. To

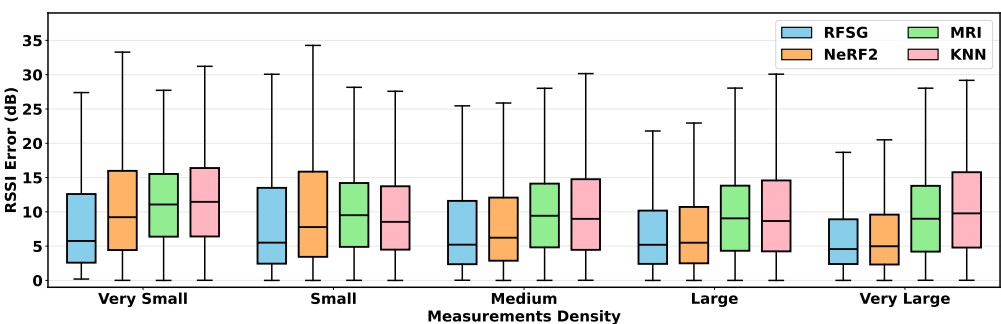

Figure 3: BLE RSSI Error (Lower values indicate better performance)

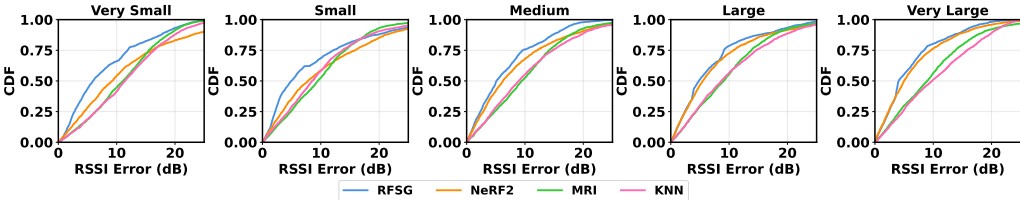

Figure 4: CDF of BLE RSSI Error

assess performance under realistic conditions, we evaluate all methods across multiple training data densities: very small (0.175 samples/sq ft), small (0.35 samples/sq ft), medium (0.70 samples/sq ft), large (∼1.50 samples/sq ft), and very large (∼3.00 samples/sq ft). The RSSI error is defined as the difference between the predicted and the collected RSSI values at all test locations.

Figures 3 and 4 summarize the comparative results under different wireless protocols. On the very small BLE RSSI dataset, RFSG reduces the median RSSI error to 5.7 dB, which is approximately at least 4 dB lower than all other baselines. As the amount of training data increases, the performance gap between ray-tracing-based models (e.g., RFSG) and ray marching-based methods (e.g., NeRF$^2$) narrows. With limited data, physics-informed models benefit from strong inductive biases that constrain the solution space, yielding more accurate predictions. However, they are also more sensitive to noise or inaccuracies in the training data. As datasets grow, ray-marching-based methods acquire sufficient information to learn approximate physical behaviors directly from the data. Their high expressive capacity enables them to capture complex mappings that were previously infeasible with tiny datasets. Furthermore, larger datasets may compensate for modeling inaccuracies in physics-based methods, further diminishing their relative advantage.

These results indicate that RFSG is more effective at learning and generalizing from limited data. This advantage stems from its strong physical constraints and well-defined stochastic geometry, which guide the model toward physically plausible solutions even when measurements are sparse.

## 6 CONCLUSIONS AND FUTURE WORK

We have presented RFSG, a novel framework that seamlessly integrates neural object representations with ray tracing for accurate RF propagation modeling. Inspired by exponential transport in stochastic geometry from computer graphics, our approach generalizes and extends these ideas to RF ray tracing, where richer and more complex interactions must be accounted for. As an advancement over physics-based ray tracing simulators widely adopted in the wireless community, RFSG holds strong potential as a versatile tool for applications such as network planning, indoor localization, and channel modeling. Despite its effectiveness, our work has several limitations that open promising avenues for future research. First, our formulation focused on the magnitude part of attenuation coefficient, while incorporating the phase component requires further exploration. Second, while we adopted exponential transport as a convenient and widely used approximation, exploring alternative non-exponential models — such as the first-passage times of Gaussian processes — represents an important direction for our further investigation.

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

## A  PROOF OF THEOREM 1

*Proof.* We begin with the key intermediate results and the corresponding notations in the proof of Theorem 4 in (Miller et al., 2024). Please refer to (Miller et al., 2024) Supplementary Material section F.1 for a more detailed illustration.

The free-space propagation distribution $p_{\boldsymbol{x},\boldsymbol{\omega}}(t)$ is the probability density that, starting from $I(\boldsymbol{x}) = 0$, the first $0 \rightarrow 1$ transition of the indicator function along the ray (that is, the first intersection) occurs at the distance $t$. For this distance to be an exponential random variable, $I_{\boldsymbol{x},\boldsymbol{\omega}}(t)$ must be a continuous-time discrete-space Markov process. The free-space propagation distance $d_{\boldsymbol{x},\boldsymbol{\omega}}^*$ (that is, the first jump time) is a exponential random variable. Let $\sigma_{01}(\boldsymbol{x},\boldsymbol{\omega})$ be the transition rate of the exponential distribution of the first jump time $d_{\boldsymbol{x},\boldsymbol{\omega}}^*$. Analogously, $\sigma_{10}(\boldsymbol{x},\boldsymbol{\omega})$ denotes the transition rate of the exponential distribution of the first jump time for the $1 \rightarrow 0$ transition. If the starting point $\boldsymbol{x}$ satisfies $I(\boldsymbol{x}) = 0$, we have:

$$\boldsymbol{\omega} \cdot \nabla v(\boldsymbol{x}) = -v(\boldsymbol{x})\sigma_{01}^{(0)}(\boldsymbol{x},\boldsymbol{\omega}) + (1 - v(\boldsymbol{x}))\sigma_{10}^{(0)}(\boldsymbol{x},\boldsymbol{\omega}), \tag{26}$$

$$-\boldsymbol{\omega} \cdot \nabla v(\boldsymbol{x}) = -v(\boldsymbol{x})\sigma_{01}^{(0)}(\boldsymbol{x},-\boldsymbol{\omega}) + (1 - v(\boldsymbol{x}))\sigma_{10}^{(0)}(\boldsymbol{x},-\boldsymbol{\omega}). \tag{27}$$

Similarly, if the starting point $\boldsymbol{x}$ satisfies $I(\boldsymbol{x}) = 1$:

$$\boldsymbol{\omega} \cdot \nabla v(\boldsymbol{x}) = -v(\boldsymbol{x})\sigma_{01}^{(1)}(\boldsymbol{x},\boldsymbol{\omega}) + (1 - v(\boldsymbol{x}))\sigma_{10}^{(1)}(\boldsymbol{x},\boldsymbol{\omega}), \tag{28}$$

$$-\boldsymbol{\omega} \cdot \nabla v(\boldsymbol{x}) = -v(\boldsymbol{x})\sigma_{01}^{(1)}(\boldsymbol{x},-\boldsymbol{\omega}) + (1 - v(\boldsymbol{x}))\sigma_{10}^{(1)}(\boldsymbol{x},-\boldsymbol{\omega}). \tag{29}$$

**Reciprocity.** We will use the free-space case as an example, the basic reciprocity for propagation within objects can be derived in the same way but different notations. Under the basic reciprocity assumption, we have $T_{\boldsymbol{x},\boldsymbol{\omega}}^{(0)}(t) = T_{\boldsymbol{y},-\boldsymbol{\omega}}^{(0)}(t)$ if $y = \boldsymbol{r}_{\boldsymbol{x},\boldsymbol{\omega}}(t)$ and $I(\boldsymbol{x}) = 0$. Then for any $\boldsymbol{\omega}$, Eq. 8 implies:

$$\int_0^t \sigma_{01}^{(0)}(\boldsymbol{r}_{\boldsymbol{x},\boldsymbol{\omega}}(s),\boldsymbol{\omega})ds = \int_0^t \sigma_{01}^{(0)}(\boldsymbol{r}_{\boldsymbol{y},-\boldsymbol{\omega}}(s),-\boldsymbol{\omega})ds. \tag{30}$$

Differentiating with respect to the distance and using the Leibniz integral rule, we have

$$\sigma_{01}^{(0)}(\boldsymbol{r}_{\boldsymbol{x},\boldsymbol{\omega}}(t),\boldsymbol{\omega}) = \sigma_{01}^{(0)}(\boldsymbol{r}_{\boldsymbol{y},-\boldsymbol{\omega}}(t),-\boldsymbol{\omega}) + \int_0^t \frac{d}{dt}\sigma_{01}^{(0)}(\boldsymbol{r}_{\boldsymbol{y},-\boldsymbol{\omega}}(s),-\boldsymbol{\omega})ds. \tag{31}$$

Note that

$$\frac{d}{dt}\sigma_{01}^{(0)}(\boldsymbol{r}_{\boldsymbol{y},-\boldsymbol{\omega}}(s),-\boldsymbol{\omega}) = -\boldsymbol{\omega} \cdot \frac{\partial}{\partial \boldsymbol{z}}\sigma_{01}^{(0)}(\boldsymbol{z},-\boldsymbol{\omega})\big|_{\boldsymbol{z}=\boldsymbol{r}_{\boldsymbol{y},-\boldsymbol{\omega}}(s)} \leq B, \tag{32}$$

where $B \in \mathbb{R}$ is a certain upper bound. Since $\sigma_{01}^{(0)}(\boldsymbol{x},\boldsymbol{\omega})$ is differentiable with respect to the point $\boldsymbol{x}$. Taking $t = 0$, we have

$$\sigma_{01}^{(0)}(\boldsymbol{x},\boldsymbol{\omega}) = \sigma_{01}^{(0)}(\boldsymbol{x},-\boldsymbol{\omega}). \tag{33}$$

Similarly, when $I(\boldsymbol{x}) = 1$, we can derive $\sigma_{10}^{(1)}(\boldsymbol{x},\boldsymbol{\omega}) = \sigma_{10}^{(1)}(\boldsymbol{x},-\boldsymbol{\omega})$ following the same logistic. Once the basic reciprocity is satisfied, Eq. 10 and Eq. 11 trivially holds.

Eq. 26, 27 and 33 form an underdetermined linear system that admits the solution

$$\begin{bmatrix} \sigma_{01}^{(0)}(\boldsymbol{x},\boldsymbol{\omega}) \\ \sigma_{01}^{(0)}(\boldsymbol{x},-\boldsymbol{\omega}) \\ \sigma_{10}^{(0)}(\boldsymbol{x},\boldsymbol{\omega}) \\ \sigma_{10}^{(0)}(\boldsymbol{x},-\boldsymbol{\omega}) \end{bmatrix} = \begin{bmatrix} \frac{|\boldsymbol{\omega}\cdot\nabla v(\boldsymbol{x})|+\tau(\boldsymbol{x},\boldsymbol{\omega})(1-v(\boldsymbol{x}))}{v(\boldsymbol{x})} \\ \frac{|\boldsymbol{\omega}\cdot\nabla v(\boldsymbol{x})|+\tau(\boldsymbol{x},\boldsymbol{\omega})(1-v(\boldsymbol{x}))}{v(\boldsymbol{x})} \\ \mathbf{1}_{\{\boldsymbol{\omega}\cdot\nabla v(\boldsymbol{x})\geq 0\}} \cdot \frac{2|\boldsymbol{\omega}\cdot\nabla v(\boldsymbol{x})|}{1-v(\boldsymbol{x})} + \tau(\boldsymbol{x},\boldsymbol{\omega}) \\ \mathbf{1}_{\{\boldsymbol{\omega}\cdot\nabla v(\boldsymbol{x})< 0\}} \cdot \frac{2|\boldsymbol{\omega}\cdot\nabla v(\boldsymbol{x})|}{1-v(\boldsymbol{x})} + \tau(\boldsymbol{x},\boldsymbol{\omega}) \end{bmatrix}, \tag{34}$$

where $\tau(\boldsymbol{x},\boldsymbol{\omega}) \geq 0$ is a free variable.

**Reversibility.** The above equation contains the undetermined function $\tau(\boldsymbol{x},\boldsymbol{\omega})$, which is not fixed by enforcing reciprocity and exponential transport alone. To identify $\tau(\boldsymbol{x},\boldsymbol{\omega})$, observe that for any two points $\boldsymbol{x}, \boldsymbol{y}$, Eq. 26 permits computing the vacancy $v$ at one point from that at the other by integrating along the straight line segment joining them. Since these vacancies represent probabilities of the same underlying random field $I$, the result must be consistent regardless of integration direction—whether from $\boldsymbol{x}$ to $\boldsymbol{y}$ or from $\boldsymbol{y}$ to $\boldsymbol{x}$. In ray notation, this bidirectional consistency implies the following constraint:
Given $\boldsymbol{x}, \boldsymbol{\omega}$, we define $\mathcal{V}_{\boldsymbol{x},\boldsymbol{\omega}}(t) \triangleq v(\boldsymbol{x} + \boldsymbol{\omega}t)$ for any $t \in \mathbb{R}$,

- In one direction, we start at $\boldsymbol{r}_{\boldsymbol{x},\boldsymbol{\omega}}(0) = \boldsymbol{r}_{\boldsymbol{x},\boldsymbol{x}\to\boldsymbol{y}}(0) = \boldsymbol{x}$ with initial condition $\mathcal{V}_{\boldsymbol{x},\boldsymbol{x}\to\boldsymbol{y}}(0) = v(\boldsymbol{x})$. Note that $\frac{d}{dt}\mathcal{V}_{\boldsymbol{x},\boldsymbol{x}\to\boldsymbol{y}}(t) = \boldsymbol{\omega}\cdot\nabla_{\boldsymbol{z}}v(\boldsymbol{z})\big|_{\boldsymbol{z}=\boldsymbol{x}+\boldsymbol{\omega}t}$. Integrating the left hand side of equation 26 with respect to $t$ along the ray until we reach $\boldsymbol{r}_{\boldsymbol{x},\boldsymbol{x}\to\boldsymbol{y}}(\|\boldsymbol{y}-\boldsymbol{x}\|) = \boldsymbol{y}$, the integration results should be $\mathcal{V}_{\boldsymbol{x},\boldsymbol{x}\to\boldsymbol{y}}(\|\boldsymbol{y}-\boldsymbol{x}\|) = v(\boldsymbol{y})$.

- Analogously, in the reverse direction, we start at $\boldsymbol{r}_{\boldsymbol{y},-\boldsymbol{\omega}}(0) = \boldsymbol{y}$ with initial condition $\mathcal{V}_{\boldsymbol{y},\boldsymbol{y}\to\boldsymbol{x}}(0) = v(\boldsymbol{y})$ and integrating the left hand side of equation 26 with respect to $t$ along the ray until we reach $\boldsymbol{r}_{\boldsymbol{y},\boldsymbol{y}\to\boldsymbol{x}}(\|\boldsymbol{y}-\boldsymbol{x}\|) = \boldsymbol{x}$. The integration results should be $\mathcal{V}_{\boldsymbol{y},\boldsymbol{y}\to\boldsymbol{x}}(\|\boldsymbol{y}-\boldsymbol{x}\|) = v(\boldsymbol{x})$.

Consider the homogeneous linear ordinary differential equation $\boldsymbol{\omega}\cdot\nabla v(\boldsymbol{x}) = g(\boldsymbol{x},\boldsymbol{\omega})\cdot v(\boldsymbol{x})$, where $g(\boldsymbol{x},\boldsymbol{\omega}) = g(\boldsymbol{x},-\boldsymbol{\omega})$. This ODE admits solutions of exponential form, i.e.

$$\underbrace{\mathcal{V}_{\boldsymbol{x},\boldsymbol{x}\to\boldsymbol{y}}(\|\boldsymbol{y}-\boldsymbol{x}\|)}_{v(\boldsymbol{y})} = \underbrace{\mathcal{V}_{\boldsymbol{x},\boldsymbol{x}\to\boldsymbol{y}}(0)}_{v(\boldsymbol{x})}\cdot\exp\left(\int_0^{\|\boldsymbol{y}-\boldsymbol{x}\|}g(\boldsymbol{x}+\boldsymbol{\omega}t,\boldsymbol{\omega})dt\right), \tag{35}$$

$$\underbrace{\mathcal{V}_{\boldsymbol{y},\boldsymbol{y}\to\boldsymbol{x}}(\|\boldsymbol{y}-\boldsymbol{x}\|)}_{v(\boldsymbol{x})} = \underbrace{\mathcal{V}_{\boldsymbol{y},\boldsymbol{y}\to\boldsymbol{x}}(0)}_{v(\boldsymbol{y})}\cdot\exp\left(\int_{\|\boldsymbol{y}-\boldsymbol{x}\|}^0 g(\boldsymbol{y}-\boldsymbol{\omega}t,-\boldsymbol{\omega})dt\right), \tag{36}$$

which satisfies the forward/backward integration for reversibility guarantee. In this case, we must have $\tau(\boldsymbol{x},\boldsymbol{\omega})\propto\frac{v(\boldsymbol{x})}{1-v(\boldsymbol{x})}$.

**Material/Medium Dependency.** Moreover, we also want to take material property into account to distinguish different attenuation magnitude degradation in different material. However, a practical object's material properties are distributed unevenly across its surface/interior and are often not measurable. Therefore, we simply take $\tau(\boldsymbol{x},\boldsymbol{\omega}) = \frac{m(\boldsymbol{x})v(\boldsymbol{x})|\boldsymbol{\omega}\cdot\nabla v(\boldsymbol{x})|}{1-v(\boldsymbol{x})}$, where $m(\boldsymbol{x})$ is a coordinate-dependent coefficient that accounts for material/medium property. In this way, we have

$$\sigma_{01}^{(0)}(\boldsymbol{x},\boldsymbol{\omega}) = \frac{(m(\boldsymbol{x})v(\boldsymbol{x})+1)|\boldsymbol{\omega}\cdot\nabla v(\boldsymbol{x})|}{v(\boldsymbol{x})}. \tag{37}$$

Analogously, starting from $I(\boldsymbol{x}) = 1$, the distance $t$ in which the first $1\to 0$ transition occurs is an exponential random variable. Enforcing reciprocity (i.e. $\sigma_{10}^{(1)}(\boldsymbol{x},\boldsymbol{\omega}) = \sigma_{10}^{(1)}(\boldsymbol{x},-\boldsymbol{\omega})$), reversibility, and material dependency, we have

$$\sigma_{10}^{(1)}(\boldsymbol{x},\boldsymbol{\omega}) = \frac{(m(\boldsymbol{x})(1-v(\boldsymbol{x}))+1)|\boldsymbol{\omega}\cdot\nabla v(\boldsymbol{x})|}{1-v(\boldsymbol{x})}. \tag{38}$$

Thus, we conclude that

$$\sigma_\delta(\boldsymbol{x},\boldsymbol{\omega}) = \begin{cases}\sigma_{01}^{(0)}(\boldsymbol{x},\boldsymbol{\omega}), & \text{if } I(\boldsymbol{x})=0\\\sigma_{10}^{(1)}(\boldsymbol{x},\boldsymbol{\omega}), & \text{if } I(\boldsymbol{x})=1\end{cases} = \begin{cases}\frac{(m(\boldsymbol{x})v(\boldsymbol{x})+1)|\boldsymbol{\omega}\cdot\nabla v(\boldsymbol{x})|}{v(\boldsymbol{x})}, & \text{if } I(\boldsymbol{x})=0\\\frac{(m(\boldsymbol{x})(1-v(\boldsymbol{x}))+1)|\boldsymbol{\omega}\cdot\nabla v(\boldsymbol{x})|}{1-v(\boldsymbol{x})}, & \text{if } I(\boldsymbol{x})=1\end{cases} \tag{39}$$

This concludes the proof. $\qquad\square$

# B APPROACH

## B.1 CHANNEL MODELS

Channel models are typically constructed either statistically, by defining distributions over channel attributes, or deterministically, using ray tracing. Statistical models fall short for applications involving positioning, sensing, and communications. Inspired by analogous techniques in computer graphics (Glassner, 1989), traditional ray tracing approaches (McKown & Hamilton, 1991) approximate electromagnetic wave propagation by modeling the interactions of individual rays with objects along their paths. These interactions may include reflection, diffraction, and penetration. While more efficient than directly solving Maxwell's equations, ray tracing still requires detailed environmental

knowledge and is often slow for prototyping. To remain computationally tractable, these methods typically rely on hard-coded, simplified physical models, such as the knife-edge approximation for diffraction (Lee, 1997). However, such abstractions often suffer from mismatches, necessitating laborious fine-tuning and calibration with real-world data. Moreover, improving their fidelity while keeping simulations efficient is challenging. Finally, because they are inherently non-differentiable, these models cannot be integrated into closed-loop design pipelines. To address these limitations, we propose a neural surrogate for physics-based wireless ray tracing in this paper.

## B.2 RAY ATTRIBUTES

We denote the $k$-th ray (among $K$ rays) at the $i$-th rendering segment as $r_k^{(i)}$. For notational simplicity, we omit sub- and superscripts in the remainder of this section. The wireless ray is characterized analogously to an optical ray (e.g., by its geometric direction). In addition to wireless channel attributes, we introduce meta-level attributes that facilitate the propagation and rendering of the eventual ray received at the receiver coordinate $x$Rx. The complete ray representation is given by

$$ r = \big( T, \ \tau, \ \varphi, \ p_0, \ \omega, \ t_s, \ t_{\mathrm{Rx}}, \ \rho_{\mathrm{Rx}}, \ \beta_{\mathrm{act}}, \ \beta_{\mathrm{Rx}} \big), $$

which can be grouped into three categories:

*(a) Wireless channel attributes.* As discussed in Section 4.3, these attributes are used to construct the wireless channel response.

*(b) Ray geometry.* We additionally encode the ray's geometric representation, which governs its propagation through the scene. The ray path is defined by

$$ p(t) = p_0 + t\omega, $$

where $p_0$ is the origin and $\omega$ is a unit vector denoting the ray direction. We focus on two particular solutions of $t$: $t_s$, where the ray intersects a surface (represented by the implicit function $f(\cdot)$ in our case), and $t_{\mathrm{Rx}}$, where the ray becomes tangential to a reception sphere centered at a receiver with radius $\rho_{\mathrm{Rx}}$.

*(c) Ray state.* To enable updates across rendering segments, we maintain two binary state variables. $\beta_{\mathrm{act}}$ indicates whether the ray remains active in the next segment (inactive rays have either reached a Rx or exited the scene boundary). $\beta_{\mathrm{Rx}}$ flags whether the ray has impinged on a reception sphere of a predefined radius.

## B.3 DETAILS OF RAY TRACING

**Ray–Scene Intersections.** For each ray, we are interested in its first interaction with the environment (e.g., the first object it encounters or impingement on the receiver). To this end, we consider the solutions to the line equation that defines the ray geometry:

$$ p(t) = p_0^{(i)} + t\omega^{(i)}. $$

We focus on two particular solutions of $t$:

*(a) Ray–Surface intersection.* The smallest $t > 0$ such that $p(t)$ lies on a surface. This is obtained by performing ray–surface intersections over all scene geometry and identifying the corresponding solution $t = t_s$, which yields the updated relay location $p_0^{(i)} + t_s \omega^{(i)}$.

*(b) Ray–Rx intersection.* We also consider positive solutions of $t$ for which the ray intersects the receiver, modeled as a sphere of radius $\rho_{\mathrm{Rx}}$. In this case, $t$ is computed as the projection of $x_{\mathrm{Rx}}$ onto the ray trajectory $p(t)$.

At the end of each ray–environment query, we thus obtain analytical estimates of the ray's first intersection with both the scene and, if applicable, the receiver.

**Ray–Surface Interaction.** If the ray $r_k^{(i)}$ (originating at $x_k^{(i)}$ and propagating along direction $\omega_k^{(i)}$) encounters an object at $x_k^{(i+1)}$ (as determined in the previous step), our objective is to characterize the outgoing ray that emanates from $x_k^{(i+1)}$. Specifically, we seek to estimate the new direction $d_k^{(i+1)}$ (e.g., whether the ray undergoes penetration or reflection) together with the associated change in gain (e.g., power attenuation or phase shift). This is a challenging problem, as it typically requires detailed

knowledge of the surface properties (e.g., material composition) as well as their electromagnetic (EM) characteristics (e.g., frequency-dependent responses).

Rather than explicitly modeling such effects, we instead learn them by associating spatial regions of the environment with material- and EM-specific attributes. Concretely, we employ a neural network that maps spatial coordinates to material coefficients $m(\cdot)$ and anisotropic parameters $\alpha(\cdot)$. In parallel, the network predicts the phase rotation $\varphi(\cdot)$ at surface interaction points. Using the proposed representation equation 19 and equation 21, we can estimate both the gain modification induced at surface intersections and the new propagation direction, which is subsequently determined following the procedure outlined in 4.3.

**Termination Check.** For certain special cases, ray tracing is terminated for a subset of rays to avoid erroneous re-reception of the same ray in subsequent iterations. Namely, when ray $k$ impinges on a Rx sphere of radius under $\rho_{\mathrm{Rx}}$ meters. For computational efficiency, we also terminate ray tracing when a ray propagates outside the region of interest (e.g., leaving the bounded environment).

## C    THE USE OF LARGE LANGUAGE MODELS (LLMS)

We used LLMs solely for linguistic polishing (e.g., wording, grammar, and clarity). No LLM assistance was used for research ideation, problem formulation, methodology design, experiments, or result interpretation. All scientific content was conceived and written by the authors.

