# OpenReview forum: "Radio Frequency Ray Tracing via Stochastic Geometry"
_ICLR.cc/2026/Conference — Submitted to ICLR 2026_

### Official Review · Reviewer_R1Fr · 2025-10-29

**Soundness:** 3
**Presentation:** 3
**Contribution:** 2
**Rating:** 4
**Confidence:** 4

**Summary:**

This paper proposes RFSG (Radio Frequency Stochastic Geometry), a physics-inspired framework that combines stochastic geometry theory with neural ray tracing for RF propagation modeling. It derives an exponential transport model for stochastic object geometries, linking attenuation to material properties and scene structure under physical constraints such as reciprocity and reversibility.

**Strengths:**

+ The paper introduces a novel perspective by extending exponential transport and stochastic geometry theory to RF propagation.

**Weaknesses:**

- The proposed theoretical foundation is elegant but difficult to achieve in practice. The derivation of exponential transport under stochastic geometry relies on strong assumptions, such as Poisson-distributed object geometry, differentiable occupancy fields, and reciprocal, stationary media, that rarely hold in real RF environments. Real-world propagation involves correlated structures, heterogeneous materials, and frequency-dependent wave effects that violate the independence and smoothness conditions required for exponential attenuation. As a result, the claimed physical guarantees, including reciprocity and reversibility, are theoretically appealing but may not be empirically realizable without substantial simplification. A more rigorous justification or numerical validation of these assumptions would strengthen the theoretical credibility of the framework.


- While the framework claims to integrate stochastic geometry theory with neural ray tracing, its implementation only partially reflects this formulation. The exponential transport equations are not directly solved or validated; instead, the method substitutes key quantities such as attenuation and material parameters with generic MLP regressors. This disconnect weakens the claimed physical grounding, as the neural networks effectively absorb unmodeled effects rather than enforcing the derived stochastic relationships. Without explicit constraints or analytical verification that the learned fields satisfy the proposed transport equations, the approach risks behaving as an empirical neural surrogate rather than a true realization of the underlying theory.


- The evaluation is narrow and lacks rigor. It is limited to a single 2.4 GHz BLE dataset and excludes key baselines such as WRF-GS. Metrics focus only on RSSI error, without assessing phase accuracy.

**Questions:**

Please see the points raised in the Weaknesses section.

---

> ### Author Response · Authors · 2025-12-02
>
> We sincerely thank the reviewer R1Fr for the time and effort in reviewing our paper. We hope the following responses can resolve your questions and concerns.
>
> (1) We agree that the assumptions used to derive the exponential transport model are a bit idealized and may not strictly satisfied in many real RF environments. However, this modeling choice is consistent with NeRF-based methods [1-3], which leverage voxel-based scene representations to capture scene impact on RF signal propagation and employ ray-marching-based rendering to achieve state-of-the-art fidelity in RF data synthesis, they all rely on a key underlying assumption: Ray propagation induces \textbf{exponential attenuation} of energy. We refer to this assumption as \textbf{exponential transport} and directly adopt it in our work.
>
> (2) In our setting, the intermediate parameters in the exponential transport model (e.g., material/object-dependent coefficients) are latent and have no direct ground-truth supervision. The only supervision comes from the final RSSI measurements. Consequently, we use MLPs to parametrize these coefficient fields, while the forward map strictly follows Eqs. (19)–(24). More specifically, the networks output local coefficients, which are then passed through the exponential transport equations to obtain attenuation along rays and, ultimately, the predicted received RF signal. Thus, the MLPs do not replace the transport model with a black-box mapping, but instantiate its unknown parameters. This is analogous to NeRF-style neural rendering [4], where density and radiance fields are not directly supervised but are learned through the volume rendering integral using only image observations. In addition, we include regularizers that enhance spatial smoothness and physically plausible behavior of the learned fields.
>
> (3) Due to the limited rebuttal period, we are not able to include all additional evaluations in this response. In the final revised version, we will expand the experimental section with more evaluation results and accompanying analysis to further substantiate the proposed framework.
>
> **References:**
>
> [1] Zhao, X., An, Z., Pan, Q., \& Yang, L. (2023, October). Nerf2: Neural radio-frequency radiance fields. In Proceedings of the 29th Annual International Conference on Mobile Computing and Networking (pp. 1-15).
>
> [2] Lu, H., Vattheuer, C., Mirzasoleiman, B., \& Abari, O. (2024). Newrf: A deep learning framework for wireless radiation field reconstruction and channel prediction. arXiv preprint arXiv:2403.03241.
>
> [3] Chen, X., Feng, Z., Sun, K., Qian, K., \& Zhang, X. (2024, November). Rfcanvas: Modeling rf channel by fusing visual priors and few-shot rf measurements. In Proceedings of the 22nd ACM Conference on Embedded Networked Sensor Systems (pp. 464-477).
>
> [4] Mildenhall, B., Srinivasan, P. P., Tancik, M., Barron, J. T., Ramamoorthi, R., \& Ng, R. (2021). Nerf: Representing scenes as neural radiance fields for view synthesis. Communications of the ACM, 65(1), 99-106.

---

### Official Review · Reviewer_UYEt · 2025-10-30

**Soundness:** 3
**Presentation:** 3
**Contribution:** 3
**Rating:** 6
**Confidence:** 3

**Summary:**

The paper addresses the challenge of modelling radio-frequency (RF) propagation in large, complex indoor/outdoor environments. The authors propose a framework called RFSG (Radio Frequency Stochastic Geometry) which:
1. Models unknown object geometry as a stochastic indicator field and invokes an exponential transport model on object-inside propagation distances;
2. Enforces physically relevant constraints such as reciprocity and reversibility in attenuation modelling;
3. ombines this stochastic geometry formulation with object-centric neural representations (MLPs) of geometry, material properties and anisotropy, which are integrated into a differentiable ray-tracing pipeline;
4. Conducts experiments on a real‐world BLE RSSI dataset, showing lower error and fewer samples required compared to neural baselines.

**Strengths:**

1. The use of stochastic geometry principles (e.g., indicator random fields, exponential transport) in RF ray-tracing is inventive, especially the grounding of attenuation in the pdf of object-inside distances. This moves beyond purely black-box neural surrogates.
2. The authors explicitly embed reciprocity, reversibility, and anisotropy in their formulation which is rare in neural channel modelling work.
3. Incorporating a neural representation of object geometry, material and anisotropy into a ray tracing scheme that handles free space and object-penetration segments is technically rich.
4. The evaluation shows the proposed method outperforms baselines on BLE RSSI prediction under sparse training data conditions. This supports the stated claims about sample efficiency.

**Weaknesses:**

1. The experiments use a BLE RSSI dataset (2.4 GHz beacon + 21 gateways in an indoor facility). RSSI is a coarse metric; the model is not validated on richer metrics (phase, full channel impulse response, mmWave or higher frequencies, mobility, multi-path delay profiles). This raises concerns about generalization to real communications/sensing settings.
2. The paper mostly addresses the magnitude (attenuation) component of the model; phase modelling is “left to future work”. However, phase and delay are critical in many RF applications (MIMO, localization).
3. The baseline set includes NeRF2, k-NN, and a simple interpolation model (MRI). But state-of-the-art RF channel modelling or hybrid physics-ML methods beyond NeRF2 may be missing — hence comparison may not fully represent current best practice.
4. Neural MLPs + differentiable ray tracing can become computationally heavy; the paper gives little analysis of runtime, memory or scalability to large outdoor environments, high frequencies, or large numbers of transmitters.

**Questions:**

1. How does the method generalize to frequencies other than BLE/2.4 GHz (e.g., sub-6 GHz cellular, mmWave, THz)? Would the exponential transport model still hold?
2. What are the runtime and memory footprints of your differentiable tracing + neural MLP system for a typical scene (e.g., your testbed)? How does this scale to larger outdoor areas or many transmitters/receivers?
3. How sensitive is performance to training sample density and scene changes (e.g., objects moved, new furniture)? Does the object-centric neural representation easily adapt or need full retraining?
4. Can you provide empirical evidence (or cite) that in real indoor/outdoor RF propagation environments the object-inside propagation distances follow an exponential distribution as assumed? What happens when the assumption fails (e.g., strong scattering, diffraction)?

---

> ### Author Response · Authors · 2025-12-02
>
> We sincerely thank the reviewer UYEt for the time and effort in reviewing our paper. We greatly appreciate the positive feedback. We hope the following responses can resolve your questions and concerns.
>
> (1) Regarding the phase term, our theoretical analysis in the current version focuses on the amplitude behavior and does not explicitly model phase, which we take it as an important direction for future theoretical work. However, in the implementation we do make the phase field learnable so that the model can capture phase-sensitive effects in practice.
>
> (2) Our framework allows initializing the mean implicit representation using a vision prior. In this sense, if the scene changes, we can modify the vision prior accordingly and only require a few additional radio-frequency (RF) measurements to quickly adapt to the new scene. Otherwise, if the vision prior is not available, we may need to retrain the model from scratch.
>
> (3) NeRF-based RF methods [1–3] typically adopt voxel-based scene representations to model how the environment affects RF propagation, and use ray-marching–style rendering to synthesize RF measurements with state-of-the-art fidelity. A key underlying assumption in these works is that ray propagation induces **exponential attenuation** of energy along each path. We follow this line of work and refer to this assumption as **exponential transport**, which we explicitly adopt as the basis of our theoretical formulation.
>
> (4) For visible light, the wavelength (on the order of $10^{-7}$ m) is much smaller than typical surface roughness, so reflections are predominantly diffuse. In contrast, RF wavelengths are on the order of $10^{-3}$ m, so most surfaces appear comparatively smooth to RF, and specular reflection tends to dominate over diffuse scattering [4]. Nevertheless, in our implementation we still include a diffuse component to better approximate real-world propagation.
>
> (5) Due to the limited rebuttal period, we are not able to include all additional evaluations in this response. In the final revised version, we will expand the experimental section with more evaluation results and accompanying analysis (including more recent baselines beyond NeRF$^2$, richer evaluation metrics, and a complexity analysis) to further substantiate the proposed framework.
>
> **References:**
>
> [1] Zhao, X., An, Z., Pan, Q., \& Yang, L. (2023, October). Nerf2: Neural radio-frequency radiance fields. In Proceedings of the 29th Annual International Conference on Mobile Computing and Networking (pp. 1-15).
>
> [2] Lu, H., Vattheuer, C., Mirzasoleiman, B., \& Abari, O. (2024). Newrf: A deep learning framework for wireless radiation field reconstruction and channel prediction. arXiv preprint arXiv:2403.03241.
>
> [3] Chen, X., Feng, Z., Sun, K., Qian, K., \& Zhang, X. (2024, November). Rfcanvas: Modeling rf channel by fusing visual priors and few-shot rf measurements. In Proceedings of the 22nd ACM Conference on Embedded Networked Sensor Systems (pp. 464-477).
>
> [4] Lu, J., Shanbhag, H., \& Al Hassanieh, H. GeRaF: Neural Geometry Reconstruction from Radio Frequency Signals. In The Thirty-ninth Annual Conference on Neural Information Processing Systems.

---

### Official Review · Reviewer_qmUN · 2025-11-01

**Soundness:** 3
**Presentation:** 2
**Contribution:** 2
**Rating:** 4
**Confidence:** 3

**Summary:**

This paper proposes RFSG(Radio Frequency Stochastic Geometry), a physics-based & differentiable framework for modeling radio-frequency propagation. Instead of relying on prev-known 3D geometry as in deterministic ray tracing, RFSG introduces a stochastic geometry formulation that models scene occupancy and attenuation as exponential random processes. The propagation field is parameterized by a neural implicit function $F_\Theta(x) \rightarrow (f, m, \alpha, \phi)$, encodes geometry, material, anisotropy, and phase.

**Strengths:**

1. Solid theoretical verification: The exponential transport theory as foundation is correct and novel. It's clear and correct to me to derive reciprocity and reversibility constraints from electromagnetic theory.
2. Physically consistent learning: Combines stochastic geometric priors with neural implicit modeling ensures energy conservation and consistent attenuation behavior.
3. pushed data-efficiency pain point: Achieves accurate RSSI prediction with an order of magnitude fewer samples than NeRF-style baselines.

**Weaknesses:**

1. Concerns on real-world deployment lacks clarification. For example, The stochastic integral formulation requires heavy Monte Carlo sampling and intricate parameter estimation, making large-scale deployment non-trivial.
2. Field modeling lasks complex-valued modeling: The method focuses on amplitude attenuation but didn't consider / reconstruct complex-valued (amplitude + phase) fields / ray-tracing like exisiting works.
3. Concerns about realism of scattering theory: The local isotropy and Markov continuity are theoretically correct, but based on my simulation experience, scattering simulation accuracy is much more sensitive than reflection/absorption, which requires designated evaluations to showcase the real-world vs simulation gap.
4. The dataset is relatively anrrow by just using one subdataset from $Nerf^2$.

**Questions:**

1. How does the computational cost scale with the number of scatterers or rays, num of obstacles and room size?
2. Any empirical observations on how scattering theory aligns with BLE dataset measurement? Any experience of real-world wireless experiments?

---

> ### Author Response · Authors · 2025-12-02
>
> We sincerely thank the reviewer qmUN for the time and effort in reviewing our paper. We hope the following responses can resolve your questions and concerns.
>
> (1) We preliminarily focus on room-level wireless applications in this work, where the propagation scale matches our current modeling and simulation budget. In this setting, our method uses a similar voxel resolution as NeRF-based RF methods [1–3], and therefore the computational time and memory usage are comparable to these approaches.
>
> (2) Regarding the phase term, our theoretical analysis in the current version focuses on the amplitude behavior and does not explicitly model phase, which we take it as an important direction for future theoretical work. However, in the implementation we do make the phase field learnable so that the model can capture phase-sensitive effects in practice.
>
> (3) For visible light, the wavelength (on the order of $10^{-7}$ m) is much smaller than typical surface roughness, so reflections are predominantly diffuse. In contrast, RF wavelengths are on the order of $10^{-3}$ m, so most surfaces appear comparatively smooth to RF, and specular reflection tends to dominate over diffuse scattering [4]. Nevertheless, in our implementation we still include a diffuse component to better approximate real-world propagation.
>
> (4) Due to the limited rebuttal period, we are not able to include all additional evaluations in this response. In the final revised version, we will expand the experimental section with more evaluation results and accompanying analysis to further substantiate the proposed framework.
>
> **References:**
>
> [1] Zhao, X., An, Z., Pan, Q., \& Yang, L. (2023, October). Nerf2: Neural radio-frequency radiance fields. In Proceedings of the 29th Annual International Conference on Mobile Computing and Networking (pp. 1-15).
>
> [2] Lu, H., Vattheuer, C., Mirzasoleiman, B., \& Abari, O. (2024). Newrf: A deep learning framework for wireless radiation field reconstruction and channel prediction. arXiv preprint arXiv:2403.03241.
>
> [3] Chen, X., Feng, Z., Sun, K., Qian, K., \& Zhang, X. (2024, November). Rfcanvas: Modeling rf channel by fusing visual priors and few-shot rf measurements. In Proceedings of the 22nd ACM Conference on Embedded Networked Sensor Systems (pp. 464-477).
>
> [4] Lu, J., Shanbhag, H., \& Al Hassanieh, H. GeRaF: Neural Geometry Reconstruction from Radio Frequency Signals. In The Thirty-ninth Annual Conference on Neural Information Processing Systems.

---

### Official Review · Reviewer_afuM · 2025-11-01

**Soundness:** 2
**Presentation:** 3
**Contribution:** 3
**Rating:** 4
**Confidence:** 3

**Summary:**

- The paper proposes a novel learnable approach to radio frequency (RD) propagation modelling i.e., at test-time to render the wireless channel (RSSI in this case) at a novel location
- The approach is in the direction of extending NeRF/volumetric rendering (specifically Miller et al., CVPR '24) to RF propagation. The main ideas of the paper involve:
  -  representing geometry of scene objects probabilistically;
  - modelling object-inside propagation using exponential transport model;
  - learning a neural field over wireless parameters (e.g., material, phase, anisotrophy)
- The approach is evaluated on the NeRF^2 dataset and results indicate the method is especially effective in learning in scenarios involving sparse measurements.

**Strengths:**

1. Physically-motivated rendering: The approach relies in a physically-constrained approach, which have previously shown to be beneficial to generalize and additionally interpret learnt parameters.
2. Special treatment of objects: Many existing works make some assumptions on the propagation environment e.g., availability of a simple mesh. However, these impose computation constraints and often become simplified. In constrast, this work models objects specifically, which lead to significant contribution due to scatter and penetration.

**Weaknesses:**

**1. Superficial Evaluation and Analysis**
- While the overall results are promising, I believe the evaluation is lacking. My specific concerns below:
- **(a) No analysis of learnt field**: A specific benefit of the approach is physical grounding wrt learnt fields over materials, attenuations, vacancy (e.g., see Fig. 1) parameters. However, no such analysis is provided. The paper would benefit tremendously by presenting these results (e.g., similar to Fig. 5 in Miller et al. 2024), since it convinces readers the model does learns non-degenerate solutions.
- **(b) Ablations**: The paper makes many design choices (e.g., laplacian regularization terms, anisotrophy terms) and an ablation would justify these choices.
- **(c) Simulated experiments**: While real-world results are helpful, I believe in this particular case, a more through simulated evaluation would make the paper stronger, since in this case one has ground-truth scene parameters that can be compared against the learnt parameters.

**2. Exponential transport model - Justification**
- The paper leverages findings of stochastic geometry perspectives of Miller et al., CVPR 2024. In their case, which involves computer graphics (optical light transport), the choice of exponential transport model is backed by plenty of literature and findings in computer graphics. However, it is largely unclear why this should hold for RF propagation.
- Furthermore, since there are specific models and literature on RF propagation through materials (see ITU 2040 "Effects of building materials and structures on radiowave propagation above about 100 MHz."). A discussion on comparing the chosen transport model with standard models would make the paper stronger.
- Additionally, it is also unclear how the model (specifically the ray tracing (Eq. 365-377)) accounts for more complex phenomenons e.g., scattering, diffraction.

**3. (Minor) Is phase learnt?**
- It is a bit unclear from the paper whether the phase is learnt. On one hand, formulations appear to indicate this is learnt (e.g., Fig. 2, Eq. 20-21). On the other hand, L478-482 also remarks "our formulation ... incorporating phase component requires further explication".

**Questions:**

My major concern is the lack of additional analysis. Importantly, that the formulation and training leads to the results by learning meaningful and interpretable parameters (e.g., occupancy). I fear that without additional investigation, the results might be accidentally due to overfitting or learning degenerate parameters. I am happy to increase my score if such analysis is added.

---

> ### Author Response · Authors · 2025-12-02
>
> We sincerely thank reviewer afuM for the time and effort in reviewing our paper. We hope the following responses can resolve your questions and concerns.
>
> (1) NeRF-based RF methods [1–3] typically adopt voxel-based scene representations to model how the environment affects RF propagation, and use ray-marching–style rendering to synthesize RF measurements with state-of-the-art fidelity. A key underlying assumption in these works is that ray propagation induces **exponential attenuation** of energy along each path. We follow this line of work and refer to this assumption as **exponential transport**, which we explicitly adopt as the basis of our theoretical formulation.
>
> (2) For visible light, the wavelength (on the order of $10^{-7}$ m) is much smaller than typical surface roughness, so reflections are predominantly diffuse. In contrast, RF wavelengths are on the order of $10^{-3}$ m, so most surfaces appear comparatively smooth to RF, and specular reflection tends to dominate over diffuse scattering [4]. Nevertheless, in our implementation we still include a diffuse component to better approximate real-world propagation.
>
> (3) Regarding the phase term, our theoretical analysis in the current version focuses on the amplitude behavior and does not explicitly model phase, which we take it as an important direction for future theoretical work. However, in the implementation we do make the phase field learnable so that the model can capture phase-sensitive effects in practice.
>
> (4) Due to the limited rebuttal period, we are not able to include all additional evaluations in this response. In the final revised version, we will expand the experimental section with more evaluation results and accompanying analysis to further substantiate the proposed framework.
>
> **References:**
>
> [1] Zhao, X., An, Z., Pan, Q., \& Yang, L. (2023, October). Nerf2: Neural radio-frequency radiance fields. In Proceedings of the 29th Annual International Conference on Mobile Computing and Networking (pp. 1-15).
>
> [2] Lu, H., Vattheuer, C., Mirzasoleiman, B., \& Abari, O. (2024). Newrf: A deep learning framework for wireless radiation field reconstruction and channel prediction. arXiv preprint arXiv:2403.03241.
>
> [3] Chen, X., Feng, Z., Sun, K., Qian, K., \& Zhang, X. (2024, November). Rfcanvas: Modeling rf channel by fusing visual priors and few-shot rf measurements. In Proceedings of the 22nd ACM Conference on Embedded Networked Sensor Systems (pp. 464-477).
>
> [4] Lu, J., Shanbhag, H., \& Al Hassanieh, H. GeRaF: Neural Geometry Reconstruction from Radio Frequency Signals. In The Thirty-ninth Annual Conference on Neural Information Processing Systems.

---

### Meta-Review · Area_Chair_kWwX · 2026-01-05

**Summary:**

C1. Experimental evaluation and analysis concerns. (a) There is no analysis of the learned field, making the evaluation superficial. (b) There are no ablation studies to support the design decisions. (c) There are no simulation experiments to verify that the proposed approach is working as intended. (d) The evaluations do not explore the accuracy of the scattering simulation. (e) The evaluation dataset choice is narrow and some relevant baselines are missing. (f) The experiments focus on a coarse RSSI metric. There is no evaluation of more challenging settings. (g) The paper does not provide details of runtime, memory or scalability.

C2. Justification of the adopted model. The paper builds on work from computer vision. RF propagation is often more complicated, with a greater chance of scattering and diffraction. The paper does not provide a compelling justification that the adopted model adequately accounts for practical challenges. Unlike other work, the paper does not consider phase. While the theoretical foundation is elegant, it relies on strong assumptions that rarely hold in practice. This suggests the need for a more rigorous experimental validation of the assumptions.

**Reviewer Concerns:**

The concerns C1 and C2 are major concerns regarding the paper and the response does not satisfactorily address them.

The response did not provide a revised paper or any supplementary experimental results. The authors acknowledged the need for further experimentation, but claimed that there was insufficient time during the rebuttal period to conduct the experimentation. As a result, it appears that the paper needs to go through a relatively major revision.

**Reviewer Scores:**

All reviewers are VERY UNLIKELY to changes their scores. The response did not include a revised paper and there were no additional experiments to augment the experimental analysis (which all reviewers viewed as inadequate to support the claims).

---

### Decision · Program_Chairs · 2026-01-26

Reject